# Immediate Effects of Isolated Lumbar Extension Resistance Exercise (ILEX) on Spine Posture and Mobility Measured with the IDIAG Spinal Mouse System

**DOI:** 10.3390/jfmk8020060

**Published:** 2023-05-08

**Authors:** Bruno Domokos, Lisa Beer, Stefanie Reuther, Christoph Raschka, Christoph Spang

**Affiliations:** 1Institute for Sports Science, University of Wuerzburg, Judenbuehlweg 11, 97082 Wuerzburg, Germany; 2Orthopedic Spine Centre Dr. Alfen, Schuererstraße 5, 97080 Wuerzburg, Germany

**Keywords:** low back pain, spine, posture, mobility, exercise, ILEX, surface scanner

## Abstract

Posture and mobility are important aspects for spinal health. In the context of low back pain, strategies to alter postural anomalies (e.g., hyper/hypolordosis, hyper/hypokyphosis) and mobility deficits (e.g., bending restrictions) have been of interest to researchers and clinicians. Machine-based isolated lumbar extension resistance exercise (ILEX) has been used successfully for rehabilitation of patients suffering from low back pain. The aim of this study was to analyse the immediate effects of ILEX on spinal posture and mobility. In this interventional cohort study, the posture and mobility measures of 33 healthy individuals (m = 17, f = 16; mean age 30.0 years) were taken using the surface-based Spinal Mouse system (IDIAG M360©, Fehraltdorf, Switzerland). Individuals performed one exercise set to full exhaustion with an ILEX-device (Powerspine, Wuerzburg, Germany) in a standardized setup, including uniform range of motion and time under tension. Scans were made immediately before and after the exercise. There was an immediate significant decrease in standing lumbar lordosis and thoracic kyphosis. No change could be observed in standing pelvic tilt. Mobility measures showed a significant decrease in the lumbar spine and an increase in the sacrum. The results show that ILEX alters spine posture and mobility in the short-term, which may benefit certain patient groups.

## 1. Introduction

In the context of low back pain, posture and mobility are often discussed as physical characteristics [1,2,3,4]. A strong body of evidence shows that mobility deficits such as reduced hamstring flexibility [5,6] and spinal malalignments such as thoracic hypo-/hyperkyphosis or lumbar hypo-/hyperlordosis are associated with pathophysiological processes and pain-related disorders in the spine (e.g., facet joint arthritis, lumbar stenosis, and lumbar degeneration) [7,8,9,10,11]. As a result of altered mechanical loadings, adverse biological responses of the human tissue can be observed, depending on the degree of severity [8]. To choose the right treatment, both clinicians and therapists rely on knowledge on how to modify these aspects.

Following technological advancements, researchers and practitioners have put increasing efforts in analysing effects of different exercises on spinal curvatures and mobility [12,13,14]. With the emergence of low-cost, non-invasive instruments, investigations of spinal characteristics have become more practical, and instant data on the present mobility and spinal alignment state can be generated [15,16,17]. Summarizing the data from a multitude of exercise programmes, Gonzalez-Galvez et al. (2019) as well as Ponzano et al. (2021) concluded in their meta-analyses that resistance and stretching exercises can have various effects on spinal curvature [13,14]. This underlines the potential of conservative active treatments to change back-pain-related malalignments and deficits.

Machine-based isolated lumbar extension resistance exercise (ILEX) is one type of exercise specifically designed to strengthen the deep paraspinal extensor muscles of the lower back and is applied frequently on patients suffering from a variety of low back pain conditions [18,19]. ILEX devices have been subject to clinical trials and training studies over at least three decades [18,19,20]. In these interventional studies (usually 9-week interventions), ILEX showed good clinical results, leading to substantial improvements in disability and pain [21,22]. Not less interestingly, patients in clinical practice often report that they sense a relief from pain and feel more comfortable in standing and dynamic movements shortly after performing the exercise. Therefore, following the treatment, we suspected that the immediate effects on posture and mobility are taking place, which have already been found in earlier studies with other exercise types [23,24].

Against this background, the aim of this study is to investigate potential immediate effects of ILEX on spine posture and mobility. To the best of our knowledge, this is the first study to analyse immediate effects of ILEX on posture and mobility.

## 2. Materials and Methods

### 2.1. Subjects

In total, 33 healthy, individual volunteers (17 males, 16 females) aged between 20 and 47 years (mean 29.9 years) participated in this interventional study (Table 1). There was a significant difference in BMI (*p* < 0.01) and a non-significant age difference (*p* = 0.149) between both sexes. In total, 30 of the 33 individuals also took part in the preceding study on reliability. Most of the participants were physiotherapists or sports scientists working in an orthopedic spine rehabilitation centre. None of them experienced notable moderate to severe back pain symptoms in the 12 months preceding the study. Participants were included if they performed the ILEX exercise on a regular basis (e.g., at least once a month), which allowed for proper selection of optimal resistance, leading to full exhaustion.

### 2.2. Study Context and Ethical Considerations

This work was a pilot study for a following research project examining temporal, long-term adaptations during ILEX in patients with specific, chronic LBP conditions such as radiculopathy and spondylolisthesis (16-weeks of intervention). The project included measurements of spinal curvature and mobility among many other parameters (e.g., ultrasound-derived muscle morphology, electromyographic parameters of muscle activity, psychosocial questionnaire-based measures, etc.).

The study was approved by the local ethics committee at Julius Maximilians University in Wuerzburg, Germany (ID: 1/2023). Written informed consent was obtained from all subjects involved in the study.

### 2.3. Scanning Device and Outcome Parameters

For measurements of posture and mobility before and after the intervention, the surface-based, electronic Spinal Mouse system (IDIAG M360©, Fehraltdorf, Switzerland) was applied [15]. The device had been used for similar purposes on LBP patients and for measurements of short-term effects before [23,24]. The device allows for measurements of thoracic curvature, lumbar curvature, and pelvic inclination in standing position as well as measures of thoracic, lumbar, and sacral mobility in maximal flexed position [15]. To ensure high reliability of measurements, a separate reliability study was conducted in advance. The intraclass correlations coefficient (ICC) measurements showed fair-to-high intrarater agreements (Table 2). Both measurements were made on the same day, and the skin markings were renewed for the second measurement. The results were comparable to an earlier study by Mannion et al. (2004), who found that intrarater reliability ranging from ICC 0.57 to 0.95 [16], and a more recent study by Demir et al. (2020), who measured an intrarater reliability ranging from 0.867 to 0.876 [17]. Furthermore, interrater reliability (ICC: 0.854–0.986) and between-day reliability (ICC: 0.843–0.984) was found to be good-to-high.

### 2.4. Scanning Procedure

Before measures were taken, spine markings were placed on the spinous processes of C7 (cervical spine) along the midline of the thoracic and lumbar spine to the median sacral crest at S3 level (sacrum) (Figure 1). The measurement procedure was performed according to the manufacturer’s guidelines in a standardized fashion [25]. For scans in standing position, the participant was instructed to stand upright with their feet shoulder-width apart in a relaxed casual position (Figure 1). Arms were hanging by the side, and the head was leveled neutrally forward. Measured values for posture in standing position included thoracic kyphosis (sTK), lumbar lordosis (sLL), and pelvic tilt (sPT). Mobility measures were taken in a maximal flexed, bent-over position with arms and head falling naturally and knees straight. Values included flexed thoracic mobility (fTM), flexed lumbar mobility (fLM), and flexed sacrum mobility, defined as the angle of inclination of the sacrum (fSM). All values in mobility represented the differences from the measures taken in standing position. Three scans were taken before and after the exercise.

### 2.5. Machine Settings (ILEX Device) and Exercise Procedure

Before the exercise was performed, the device (Powerspine Back, nr. 30000-367, Powerspine, Wuerzburg, Germany) (Figure 2) was electronically adjusted to the anthropometric properties of the individuals by the therapist. Settings included seat height, foot board, knee pad, and thigh belt. The participants were fixated in a semi-sitting position, ensuring biomechanics that allowed ideal activation of the lumbar extensor muscles and a reduced activation of hip and lower limb muscles. Altogether, these features represented a pelvic restraint system, which was specific to the device [18]. Finally, the counterweight was calculated to reach uniform resistance during the movement. For this study, individuals were asked to perform one set with 12 to 15 repetitions to muscle failure. As the participants were familiar with the exercise, individual resistance could be estimated precisely. Exercise conduct was guided by a visual panel, with each repetition lasting for exactly 10 s (flexion-extension cycle), providing for a standardized time under tension (TUT) between 120 and 150 s until muscle failure was reached. All participants performed the exercise in a standardized range of motion from 12° (extension) to 42° (flexion), which was pre-set in the device’s software (Figure 3). To determine effects of ILEX training on posture and mobility, measurements were taken directly, before and after the intervention. In total, the time span between pre- and post-measures did not exceed ten min.

### 2.6. Statistical Analysis

For statistical analysis, SPSS software was used (SPSS, Chicago, IL, USA). Normal distribution was tested using the Kolmogorov–Smirnov test. Parametric tests (paired/unpaired *t*-tests) were used to analyse interventional effects and differences in the reliability study. Intraclass correlation coefficient (ICC) was determined for assessment of reliability (see above). Pearson correlation coefficient^®^ was used to determine correlations between the different areas of the spine. Significance level was set to *p*-value < 0.05. Figures were made using Graphpad Prism Software (Version 9) (GraphPad Software, Boston, MA, USA).

## 3. Results

There was a decrease in sLL and sTK immediately after the ILEX intervention (Figure 4). Angle values for sLL decreased from 40.21° ± 10.31° (pre) to 38.81° ± 10.54° (post) (*p* < 0.001). Values for sTK changed from −28.97° ± 7.93° (pre) to −26.93° ± 8.03° (post) (*p* < 0.001). No significant changes could be observed for sPT (14.97° ± 6.40° to 14.63° ± 6.0°; *p* = 0.271).

Concerning mobility, a significant increase after ILEX intervention was measured for fSM (58.14° ± 12.92°(pre) to 62.25° ± 12.13° (post); *p* < 0.001) (Figure 5). fLM decreased from 58.14° ± 8.87° to 56.33° ± 8.32° (*p* < 0.01). No change in fTM could be observed (18.57° ± 7.26° (pre) to 19.40° ± 7.35° (post); *p* = 0.127). *** *p* < 0.001, ns = not significant.

There was a correlation in standing posture between the neighbouring spine areas after the ILEX exercise intervention (Table 3).

## 4. Discussion

The aim of this study was to analyze immediate effects of one set of heavy-loaded ILEX on spine posture and mobility with the Spinal Mouse. The study yielded four interesting findings: two immediate changes were measured for postural configuration, shown as lower standing lumbar lordosis (sLL) and lower standing thoracic kyphosis (sTK). The other immediate changes related to mobility, displayed as lower flexed lumbar mobility (fLM) and higher sacral mobility (fSM). Researchers have been investigating the immediate effects of exercises with the purpose to understand and find mechanisms that positively influence biophysical aspects of the body for years [23,24,26,27,28,29,30,31]. Without treatment, anomalies such as hyperkyphosis, hypolordosis, and mobility deficits can promote pathophysiological processes, leading to musculoskeletal pain conditions (e.g., low back pain) [32,33].

### 4.1. Immediate Effects on Posture

The immediate reduction in lumbar (sLL) and thoracic (sTK) standing angles found in this study resulted in a more upright standing position. The correlation analysis revealed a clear interdependence between the areas of the spine, which signified a change in biomechanics and in the mechanical load of the spine [32]. Being the central pillar of the body, the spine distributes the body’s load during movement and daily activities [34]. Deviations in lordosis or kyphosis angles may, therefore, increase or decrease shear forces and pressure on disks and vertebrae. In the short-term, the effect of a more upright standing position may explain why patients often sense a relief from pain immediately after the exercise. However, more research needs to be performed to validate this finding and to understand the exact physiological processes. In the long-term, biomechanical deficits in posture must not be ignored, as they can support the development of severe musculoskeletal disorders [2,35]. However, it should be considered that there are certain patient groups for which a reduction in lumbar lordosis is not desired (e.g., patients with anterior derangement or hypolordosis). When applied with patients, normal values (e.g., Cobb angle) should be considered before treatment [8]. The results of this study contrasted with Lopez-Minaro et al. (2012), who did not measure any immediate changes in posture with the Spinal Mouse, following a four-set hamstring-stretching programme [23].

With regard to the finding of a lower standing lumbar lordosis (sLL), Takihara et al. (2009) concluded that a lower sLL could be the result of muscle fatigue. In their Spinal Mouse study, individuals performed three sets to exhaustion of a repetitive prone back-extension exercise (45° to 0°). After the intervention, lumbar curvature significantly decreased [24]. In another study, Malai et al. (2015) analyzed the immediate effect of a hold–relax stretching protocol for the iliopsoas in patients suffering from chronic non-specific LBP with hyperlordosis. They found that patients not only showed lower SLL comparable to this study immediately after the stretching exercises but also reported a decrease in pain sentiment [26]. From a biomechanical standpoint, findings like these suggest that certain risk groups (e.g., patients with hyperlordosis) could benefit from ILEX. Yet, the result could also be a symptom of muscle fatigue, as other studies suggested.

Concerning the result of a lower standing thoracic kyphosis (sTK), a lot of research was carried out to understand mechanisms that can reverse or slow down hyperkyphosis and its progression [13,14]. The prevalence of hyperkyphosis is estimated to be around 20 to 40 percent in older adults and further increases with age [14]. It is regarded as problematic for several reasons including the risk of disc degeneration, mobility deficits, and overall reduction in quality of life [14]. Among conservative treatments analyzing immediate effects, Koo et al. (2022) found a positive postural adaptation in patients with forward shoulder posture (characterized by hyperkyphosis), following a five-set reverse plank protocol [27]. Other strategies with proven immediate corrective effects include taping [36] and the use of ortheses [28]. The effect found after one set of ILEX could, therefore, complement these strategies and benefit patient groups with hyperkyphosis.

### 4.2. Immediate Effects on Mobility

Regarding immediate changes in mobility, there was a contrasting finding in this study. While mobility increased in the sacrum (fSM), mobility in the lumbar region (fLM) was reduced. Due to their interdependence, lower limb and spinal mobility have both been acknowledged as important characteristics of a healthy spine [1,6]. Restrictions in lumbar mobility are problematic, as they lead to bending stress, expressed as excessive strain on the passive structures (e.g., disks and ligaments) [1,33]. Simultaneously, it is important that the motion scope between individual segments remains limited in a so-called *neutral zone* so that the passive structures, including the spinal column, are not at risk of damage through overstrain [1].

With regard to the lower flexed lumbar mobility (fLM), a possible explanation for lower fLM found in this study could be a higher muscle tone and blood flow. This effect could prevent exceeding segmental movement in the lumbar spine during forward bending and protect the passive structures from excessive stress. Studies including ultrasound-based measurements have shown that an increase in muscle thickness can accompany changes in kinematics [27]. Other studies with ILEX and ultrasound are underway to validate if a higher immediate muscle thickness of the multifidus could be the cause. In another interesting study, Shum et al. (2013) found higher lumbar bending mobility (active flexion range of motion) immediately after three cycles of posteroanterior mobilization, which was accompanied by a spontaneous reduction in lumbar pain [34]. It cannot be generalized, therefore, whether stiffness or flexibility in the lumbar spine is the desired state for patients with back pain.

Concerning the finding of a higher flexed sacral mobility (fSM), there is strong evidence that a restriction in hamstring mobility supports the development of several back-pain-related issues [6,33]. Due to their anatomic attachment to the pelvis, the tilting during forward bending is hindered. This study found that one set of ILEX immediately increases fSM, which was also found after strategies such as stretching [23,35] and self-myofascial release on the plantar surface (among other interventions) [36]. One explanation for this immediate change after ILEX could be that the pelvic restraint system is responsible for a reduction in tension and stiffness of the muscle–tendon units in the gluteal and hamstring areas since it does not allow for any movement of the pelvis and the lower limbs. Plus, with four seconds, the eccentric, muscle-lengthening phase is considerably long compared to usual contraction cycles. However, more research needs to be performed to validate this hypothesis.

This study was subject to several limitations. First, the results from the study group could not be simply transferred to real patients, as we only used healthy individuals. Notably, even though patients did not report any back pain issues, it could not be implied that these patients were characterized by full musculoskeletal health. It could be suggested that future studies also use imaging techniques to obtain information on the biology of the study group. Additionally, sex differences were not considered since the direction of potential biomechanical effects was the main focus of this study. Future research that specifically focuses on sex differences should, therefore, also take into account additional factors potentially influencing the results (e.g., BMI differences between male and female subgroups). Second, the analysis with a surface scanner such as the Spinal Mouse (IDIAG M360) only provided a mechanistic outside view of posture and mobility. It did not provide any information about other aspects relevant to posture- and mobility like neural activation or (subjective) stretch tolerance. Third, the main interest of this study was the immediate effects. This did not automatically allow conclusions about long-term effects on posture and mobility. For both aspects, other processes need to be considered that only played a minor role in this short-term intervention (e.g., structural adaptions of bones and tendons, muscle hypertrophy, etc.). Longitudinal studies with ILEX interventions lasting for several weeks could provide more information on these topics. 

## 5. Conclusions

First, this study has provided information on the immediate effects of heavy-loaded ILEX on posture and mobility in healthy subjects. Future studies should validate these findings in long-term, interventional studies, preferably with patients suffering from LBP or other conditions related to posture and mobility. Deriving from the study’s findings, we see many advantages of ILEX to be considered for spinal health as well as for the treatment and prevention of many musculoskeletal conditions. Future studies should also investigate other aspects relevant to posture- and mobility-like aspects of muscle morphology and activity, function, and strength.

## Figures and Tables

**Figure 1 jfmk-08-00060-f001:**
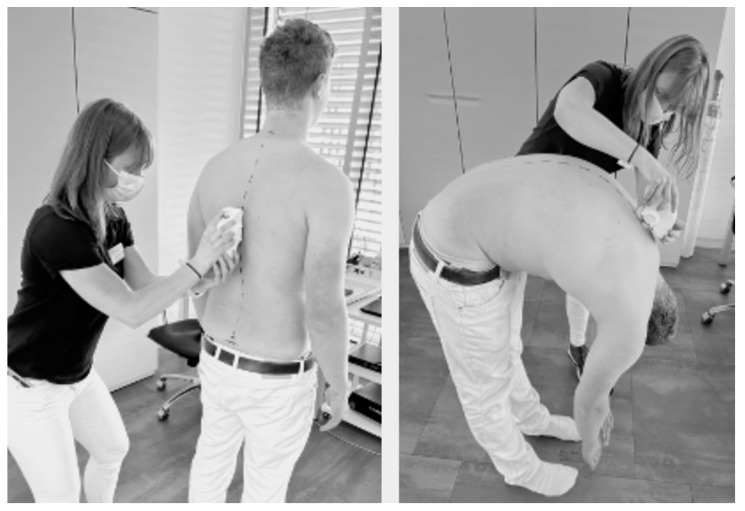
Scanning procedure in standing position (**left**) and maximal flexed, bent-over position (**right**).

**Figure 2 jfmk-08-00060-f002:**
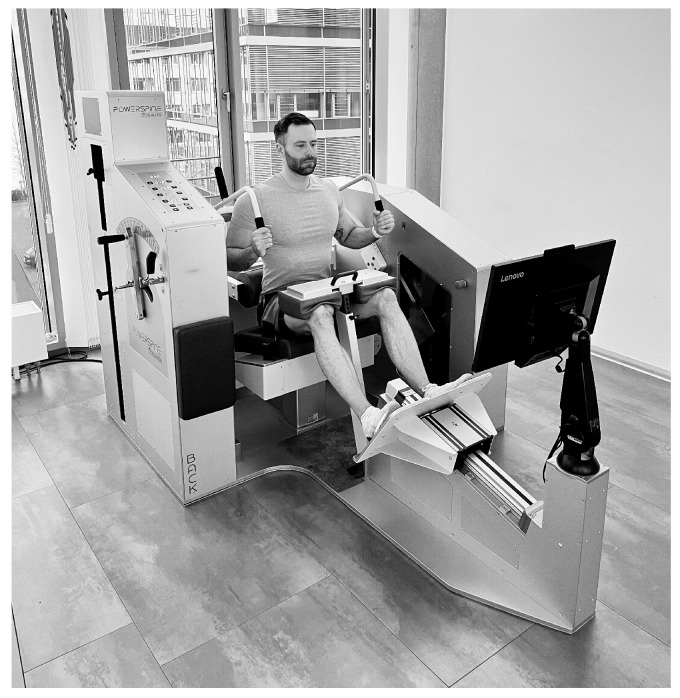
Isolated lumbar extension resistance exercise (ILEX).

**Figure 3 jfmk-08-00060-f003:**
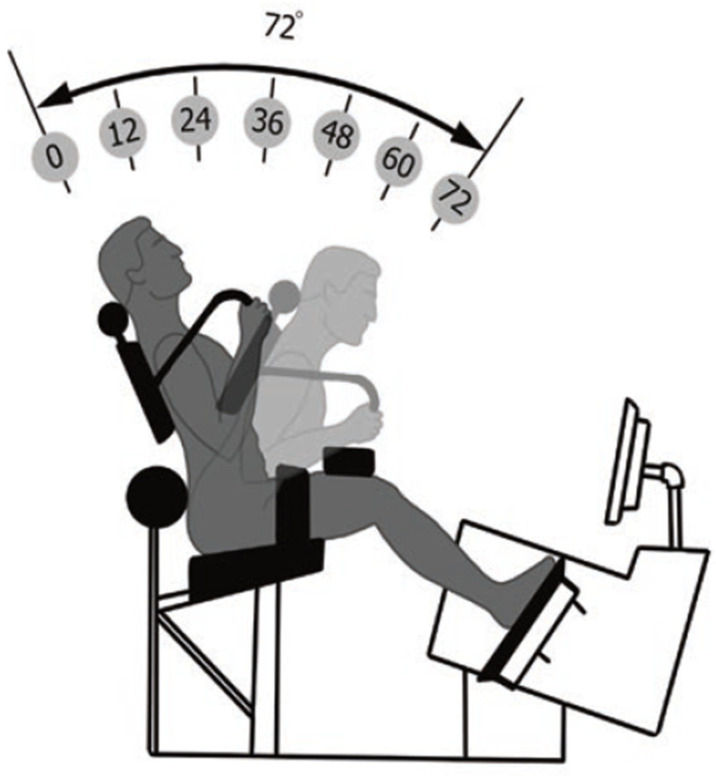
Range of motion (ROM) during the flexion-extension cycle with ILEX. In this study, ROM was restricted (12° to 42°).

**Figure 4 jfmk-08-00060-f004:**
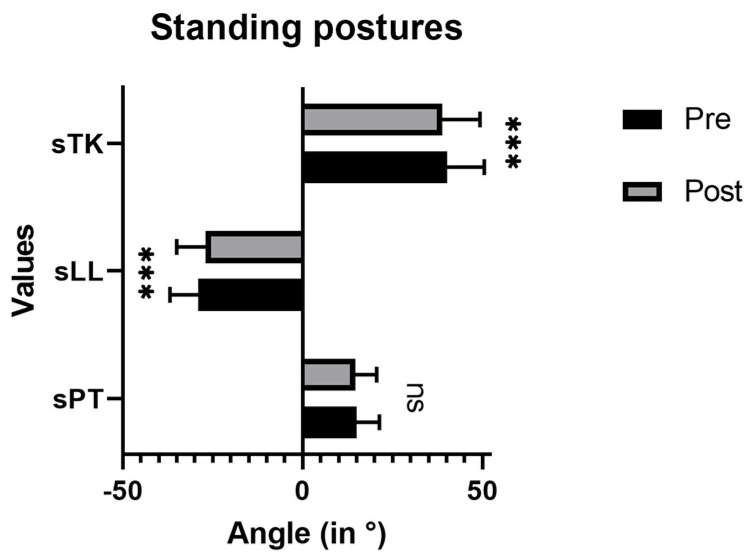
Measured angle values before and after ILEX intervention for standing postures (thoracic spine sTK, lumbar spine sLL, pelvic tilt sPT). *** *p* < 0.001, ns = not significant.

**Figure 5 jfmk-08-00060-f005:**
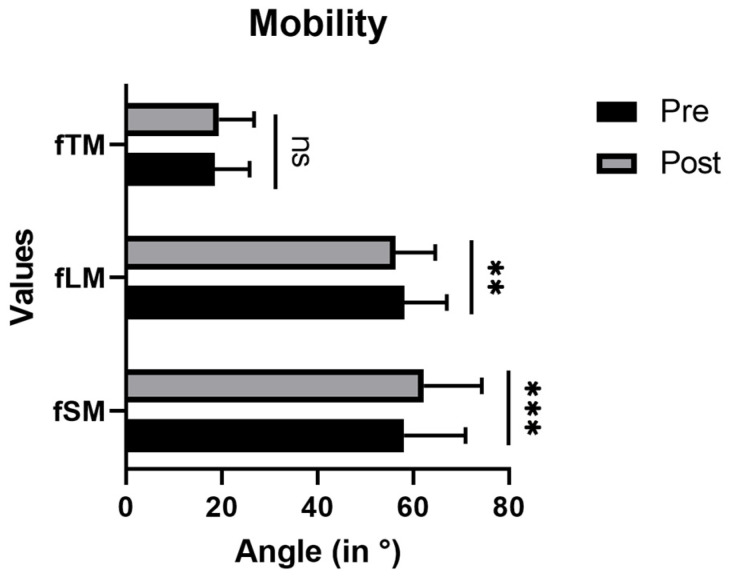
Measured angles values before and after ILEX intervention for flexed mobility (thoracic spine fTM, lumbar spine fLM, sacrum fSM). Angle values defined as differences from standing position. ** *p* < 0.01, *** *p* < 0.001, ns = not significant.

**Table 1 jfmk-08-00060-t001:** Participant characteristics.

	Women	Men	Total
*n*	16	17	33
Age (years)	28.2 ± 8.3	31.6 ± 4.6	29.9 ± 6.8
Height (cm)	167.6 ± 7.4	183.9 ± 4.6	176.0 ± 10.2
Weight (kg)	59.7 ± 6.2	83.4 ± 12.8	71.9 ± 15.6
BMI (kg/m^2^)	21.3 ± 1.9	24.7 ± 4.1	23.0 ± 3.6

**Table 2 jfmk-08-00060-t002:** Intrarater reliability and comparisons of means (paired *t*-test). ICC = intraclass correlation coefficient, SD = standard deviation, SEM standard error mean. * *p* < 0.05.

	Mean ± SD in °		*p*	ICC (95% CI)	SEM (95% CI)
	M1	M2			
Standing Postition					
sTK	40.82 ± 10.43	40.26 ± 10.56	0.342	0.977 (0.951–0.989)	0.581 (−0.627–1.749)
sLL	−28.18 ± 7.86	−26.42 ± 13.54	0.359	0.722 (0.419–0.867)	1.887 (−5.620–2.100)
sPT	14.85 ± 5.94	15.19 ± 6.66	0.372	0.974 (0.946–0.988)	0.369 (−1.088–0.420)
Mobility					
fTM	17.11 ± 7.33	17.21 ± 7.72	0.912	0.883 (0.753–0.944)	0.901 (−1.944–1.742)
fLM	56.71 ± 8.14	56.85 ± 8.34	0.726	0.983 (0.964–0.992)	0.396 (−0.949–0.669)
fSM	58.62 ± 16.23	62.43 ± 12.81	0.040 *	0.863 (0.703–0.936)	1.774 (−7.438–−0.182)

**Table 3 jfmk-08-00060-t003:** Cross table of Pearson correlation between different spine areas for standing posture and flexed mobility. * *p* < 0.05, *** *p* < 0.001.

**Standing Posture**			
	sTK	sLL	sPT
sTK		−0.385 *	0.089
sLL			−0.826 ***
**Mobility**			
	fTM	fLM	fSM
fTM		−0.169	0.349
fLM			−0.287

## Data Availability

All data are available from the corresponding author on reasonable request.

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
