# Peer review of "Immediate Effects of Isolated Lumbar Extension Resistance Exercise (ILEX) on Spine Posture and Mobility Measured with the IDIAG Spinal Mouse System"

_jfmk, 2023, doi:10.3390/jfmk8020060_

Round 1
Reviewer 1 Report
Dear Author
This research is a valuable article that demonstrates the acute effects of ILEX on the spine using a method that has gained credibility.
On the other hand, there are a number of points that need to be revised, see below.
Major points
l There were differences in age and BMI between men and women in the participants in this study, and it is recommended that these be compared in the statistical analysis. If there are differences in age and BMI between men and women, this should be taken into account in the discussion.
l As Table 4) is the main finding in this study, it should be shown in Fig.
l
Mainor points
・As both strength exercise and resistance exercise are used in this paper, a choice should be made between them.
・Table1(in yesrs), (in cm), (in kg)→in deleted.
・BMI→(㎏/m2)
・2.2. Sudy context and ethical considerations→Study
Reviewer 2 Report
The term acute effects in the title and content of the article seems inappropriate, as the word acute refers to systemic pathologies rather than to the effects of exercise. The term immediate effects would be more appropriate.
It is advisable to describe more extensively in the introduction the pathomechanisms of pain syndromes depending on the degree of severity.
The effects obtained in the form of reduced spinal curvature are unlikely to be beneficial for all patients. They may be beneficial for patients with anterior derengement according to McKenzie (anterior disc protrusion), but such patients are only a few per cent among LBS patients.
The described exercises may also be useful in persons with lumbar hypermobility and a deepened lordosis of the lumbar spine, a group that should be closely matched. It is not known whether the study group was subject to such a classification.
It is difficult to consider the subjects as healthy when they had not suffered from LBP during the year before the study. The absence of pain at any given time does not imply full musculoskeletal health. We did not receive full information about the subjects' pain (e.g. pain scales at the time before and after the study). It was not determined whether they experienced pain during the study. Conducting a motor examination to exhaustion without imaging documentation seems inappropriate.
It is not known how the reduction in spinal curvature correlated with any pain in the subjects, was it present, did it decrease or centralise?
The norms of angular value for lumbar lordosis that were considered normal are not given, so it is not known whether a reduction in lordosis was desirable, and the same is true for of spinal mobility.
The sacrum is a bony prominence and is immobile. Did the authors mean mobility of the lumbosacral junction? Excessive mobility of the L5/S1 motion segment leads to disc wear at this level and pathology. It is not known whether increasing mobility in this area is beneficial in tested persons.
A short observation in the examined range is unreliable. It should be checked what effects the described study had in the longer term, even up to a month. Particularly with regard to the possibility of pain.
The selection of participants from among the employees of the centre where the authors of the study are affiliated does not seem to be appropriate for ethical reasons, due to possible work relationships. It should be clarified whether there was a professional relationship between the subjects and the authors of the study.
The IDIAG Spinal Mouse system seems to be a good non-invasive measurement tool of the scope necessary for the study.
Round 2
Reviewer 1 Report
Dear Author.
Thank you very much for revising your paper in response to my remarks. There are no further remarks from me. I look forward to further development of your research.
Author Response
Thank you very much!
Reviewer 2 Report
After the changes made by the authors, the article is eligible for publication. However, the angle of the lumbosacral flexion and the angular values of the depth of the lumbar lordosis and thoracic kyphosis have the norms given in the literature. For example, the value of this angle for lumbar lordosis is ± 135-140°. The more flattened the lumbar lordosis, the greater the value of lumbosacral angle. Nevertheless, in postural defects or other pathologies, shallow lordosis may be accompanied by a horizontal arrangement of the sacrum. That is, the angular value of the L5/S1 junction ( ± 125°) is lower with the increased angular value of the lumbar lordosis, and this is not a biomechanically favorable situation. Therefore, in the description of the figures in the article, norms should be given, with an indication that only differences in mobility were assessed. In the continuation of the study, the above angular values should be determined radiologically during the examination.
Other than that, I have no other comments
Author Response
Dear Reviewer,
thank you very much for this final comment. After in-depth discussion in our team, we have decided not to include normal, absolute values in this study with healthy individuals, but to include this information in our larger, ongoing study with LBP patients. We followed your suggestion to specify that only differences in mobility were assessed in the Materials section as well as in the Figure legend.
Thank you again for your valuable insights, which have improved our manuscript in many aspects.
Kind regards,